# Metformin and Androgen Receptor-Axis-Targeted (ARAT) Agents Induce Two PARP-1-Dependent Cell Death Pathways in Androgen-Sensitive Human Prostate Cancer Cells

**DOI:** 10.3390/cancers13040633

**Published:** 2021-02-05

**Authors:** Yi Xie, Linbo Wang, Mohammad A. Khan, Anne W. Hamburger, Wei Guang, Antonino Passaniti, Kashif Munir, Douglas D. Ross, Michael Dean, Arif Hussain

**Affiliations:** 1Greenebaum Comprehensive Cancer Center, University of Maryland, Baltimore, MD 21201, USA; LiWang@som.umaryland.edu (L.W.); makhan@som.umaryland.edu (M.A.K.); ahamburg@som.umaryland.edu (A.W.H.); WGuang@som.umaryland.edu (W.G.); tpassaniti@som.umaryland.edu (A.P.); ddross@som.umaryland.edu (D.D.R.); 2Department of Pathology, University of Maryland School of Medicine, Baltimore, MD 21201, USA; 3Baltimore VA Medical Center, Baltimore, MD 21201, USA; 4Division of Endocrinology, University of Maryland School of Medicine, Baltimore, MD 21201, USA; kmunir@som.umaryland.edu; 5Department of Medicine, University of Maryland School of Medicine, Baltimore, MD 21201, USA; 6Division of Cancer Epidemiology and Genetics, National Cancer Institute, Bethesda, MD 20892, USA; deanm@mail.nih.gov; 7Department of Molecular Biology and Biochemistry, University of Maryland School of Medicine, Baltimore, MD 21210, USA

**Keywords:** prostate cancer, metformin, ARAT, PARP-1, poly(ADP-ribose) (PAR), lysosome

## Abstract

**Simple Summary:**

In the present study, we sought to determine whether a commonly used oral drug to treat adult-onset diabetes, metformin, which has a longstanding clinical history and known safety and tolerability profile, can improve the anti-cancer effects of two well-established oral agents currently in use to treat advanced prostate cancer, abiraterone and enzalutamide. We used androgen-sensitive cell culture models of human prostate cancer to test our hypothesis. We found that metformin and the oral anti-prostate cancer agents together are more effective in inhibiting prostate cancer cell growth and inducing prostate cancer cell death than when used alone. We identified new pathways by which the enhanced anti-cancer effects occur with the combination treatments. The present work suggests that incorporating metformin with abiraterone or enzalutamide may improve treatment outcomes in hormone sensitive prostate cancer.

**Abstract:**

We explored whether the anti-prostate cancer (PC) activity of the androgen receptor-axis-targeted agents (ARATs) abiraterone and enzalutamide is enhanced by metformin. Using complementary biological and molecular approaches, we determined the associated underlying mechanisms in pre-clinical androgen-sensitive PC models. ARATs increased androgren receptors (ARs) in LNCaP and AR/ARv7 (AR variant) in VCaP cells, inhibited cell proliferation in both, and induced poly(ADP-ribose) polymerase-1 (PARP-1) cleavage and death in VCaP but not LNCaP cells. Metformin decreased AR and ARv7 expression and induced cleaved PARP-1-associated death in both cell lines. Metformin with abiraterone or enzalutamide decreased AR and ARv7 expression showed greater inhibition of cell proliferation and greater induction of cell death than single agent treatments. Combination treatments led to increased cleaved PARP-1 and enhanced PARP-1 activity manifested by increases in poly(ADP-ribose) (PAR) and nuclear accumulation of apoptosis inducing factor (AIF). Enhanced annexin V staining occurred in LNCaP cells only with metformin/ARAT combinations, but no caspase 3 recruitment occurred in either cell line. Finally, metformin and metformin/ARAT combinations increased lysosomal permeability resulting in cathepsin G-mediated PARP-1 cleavage and cell death. In conclusion, metformin enhances the efficacy of abiraterone and enzalutamide via two PARP-1-dependent, caspase 3-independent pathways, providing a rationale to evaluate these combinations in castration-sensitive PC.

## 1. Introduction

Prostate cancer (PC) is the second leading cause of cancer-related deaths among men in Western countries [1]. A mainstay of treating patients with advanced PC is androgen deprivation therapy (ADT), which involves removal of gonadal sources of testosterone via surgical (bilateral orchiectomy) or medical castration (LHRH agonists or antagonists). ADT induces initial responses in the majority of patients by disrupting androgen receptor (AR)-axis signaling. However, the disease eventually progresses within two years of ADT in most patients despite castrate levels of serum testosterone, resulting in castration resistant prostate cancer (CRPC) [2]. Such progression is often due to the restoration of androgen-AR signaling under androgen deprived conditions. Therefore, agents targeting either androgen biosynthesis (e.g., abiraterone acetate (Abi)) or AR signaling (e.g., enzalutamide (Enz)), i.e., the so-called androgen receptor-axis-targeted (ARAT) agents, were introduced as a second line therapy in patients with CRPC [3]. While both Abi and Enz can improve overall survival among responding men, these treatments also eventually fail, resulting in disease progression [4,5,6,7]. Further, other men with CRPC may be resistant to Abi or Enz de novo [8,9]. Studies suggest that resistance to Enz and Abi may in part be due to altered expression of AR and/or AR splice variants in PC cells [10]. 

The finding that patients with diabetes taking metformin, but not other anti-diabetic drugs, have a decreased risk of dying from PC [11,12] led to extensive studies on the anti-tumor effects of metformin in PC. Pre-clinical studies demonstrate that metformin can down-regulate AR by disrupting the protein midline-1 (MID1) complex, which otherwise increases AR via enhanced translation [13]. Other studies demonstrate that metformin can induce apoptotic cell death [14,15,16,17]. However, as the pharmacologic concentrations used in most of these studies are not readily achievable clinically, the beneficial effects of metformin have been difficult to ascertain in patients with PC.

In addition to caspase-dependent apoptosis, alternative models of programmed cell death (PCD) have been proposed [18,19,20]. Apoptotic cell death can proceed through the activation of both caspase-dependent and -independent pathways. Caspase-independent PCD pathways are important when caspase-mediated routes fail. For caspase-independent PCD, poly (ADP-ribose) polymerase-1 (PARP-1) plays a central role. Normally PARP-1 is involved in the repair of DNA damage induced by a variety of cellular stresses. However, additional functions of PARP-1 have also been revealed (for review, see [21,22,23]). PARP-1 can be cleaved by several ‘suicidal’ proteases. The cleaved PARP-1 fragment containing a DNA binding domain can still bind to DNA but cannot catalyze DNA repair as it lacks the catalytic domain. Thus, the cleaved PARP-1 fragment that binds DNA can act as a dominant-negative inhibitor of PARP-1, inhibiting DNA repair and leading to cell death.

Excessive activation of PARP-1 can also lead to a 10–500-fold increase in poly ADP ribose (PAR) polymer accumulation within the nucleus [24], which then translocates to the mitochondria to cause a release of mitochondrial apoptosis inducing factor (AIF). AIF release and its subsequent translocation to the nucleus can commit cells to undergo parthanatos [25]. This phenomenon was first described in neuronal cells undergoing neural degradation and has also been linked to other syndromes connected with specific tissue damage [26,27,28].

In this paper, we used two human androgen-sensitive PC cell lines, LNCaP and VCaP, to study the role of metformin and ARATs in prostate cancer. We report that in PC cells, metformin, in combination with an inhibitor of androgen biosynthesis (Abi) or an AR targeting agent (Enz), can mediate PARP-1-dependent PCD: a) via enhanced PARP-1 cleavage that is essentially independent of caspase 3 activation, b) via enhanced PARP-1 activation, and c) at lower concentrations than have been observed with metformin in prior studies. This report expands the possible pathways by which metformin-based and ARAT-based targeting strategies could potentially be further developed and enhanced to treat PC.

## 2. Results

### 2.1. Effects of Enz and Abi on LNCaP and VCaP Cells

#### 2.1.1. Response of LNCaP and VCaP Cells to Enz or Abi—Effects on Cell Proliferation and Cell Death

Human androgen-sensitive PC cell lines LNCaP and VCaP were treated with increasing concentrations of Enz or Abi (range 0.3 to 80 µM) for 3 days. Cell proliferation was determined by 3-(4,5-dimethylthiazol-2-yl)-2,5-diphenyltetrazolium bromide (MTT) assay. Enz or Abi treatment resulted in inhibition of cell proliferation (statistically significant inhibition begins to occur at 0.3 µM Enz or Abi in LNCaP cells (Figure 1A) and 2.5 µM Enz or Abi in VCaP cells, respectively (Figure 1B)), with a further dose-dependent increase in growth inhibition occurring at doses >20 µM for LNCaP cells and doses >40 µM for VCaP cells.

As proof of principle to understand the effects of ARATs (and metformin—see below) in the PC cells, and to make the study less cumbersome, we evaluated fixed doses of Abi and Enz for both cell lines in several experiments. For LNCaP cells, the fixed doses used were close to the respective IC50 values for these drugs. For VCaP cells, although the drug doses tested were lower than the IC50 values, the Abi and Enz concentrations were in the range of what has been used by other investigators for these cell lines [29,30]. Both LNCaP and VCaP cells have long cell population doubling times (42 to 51+ h, respectively) so that any manifestations of anti-tumor responses in these cells are likely to occur after longer drug exposures compared to what might be observed with highly toxic agents against more rapidly dividing cells. Although in vitro experiments do not necessarily recapitulate in vivo effects, looking at outcomes after relatively longer durations of drug treatment in vitro is also not inconsistent from the perspective of hormonal therapy in patients with PC who are treated for several weeks with these agents before assessing treatment outcomes. With these considerations in mind, we chose to assess treatment responses after 5 days drug exposures in several studies presented below.

Trypan blue staining was performed to determine whether there was any associated component of cell death amongst the treated cells. We tested fixed doses of Enz and Abi that are closer to clinically achievable concentrations in patient serum [31,32] and also within range of the respective IC50 values for LNCaP cells or used by other investigators for VCaP cells (Appendix A). LNCaP or VCaP cells were treated with 10 µM Enz or 5 µM Abi for 5 days, then both floating and attached cells were collected and stained with trypan blue. At these doses, neither Enz nor Abi induced cell death in LNCaP cells as no increase in floating trypan blue positive dead cells occurred (Figure 1C left panel), although they did inhibit cell proliferation (attached, mostly trypan blue negative) (Figure 1C right panel). By contrast, in VCaP cells, both Enz (10 µM) and Abi (5 µM) not only reduced the total number of attached cells (Figure 1D right panel) but also increased, albeit small, the proportion of floating trypan blue positive dead cells (5–7% of total cells in the treated population compared to 0.5% in control cells (Figure 1D left panel)). Taken together, these data indicate that at the concentrations tested, Abi and Enz inhibit cell proliferation in both LNCaP and VCaP cells and induced some floating dead cells among the ARAT-treated VCaP cells.

#### 2.1.2. Effects of Enz and Abi on Prostate Serum Antigen (PSA), AR, AR-v7 Expression and PARP-1 Cleavage in LNCaP and VCaP Cells

Consistent with prior studies, Western blot analysis reveals that treatment with Abi (5 µM) or Enz (10 µM) for 5 days results in an approximately 1.5-fold increase in AR expression in LNCaP cells (Figure 1E left panel, Appendix A), whereas the expression of the AR target gene PSA is decreased. In contrast to LNCaP cells, VCaP cells express both AR and ARv7 but minimal basal levels of PSA. Enz or Abi treatment for 5 days primarily increased protein levels of ARv7 in the VCaP cells (Figure 1E right panel). These changes in AR or ARv7 protein levels in the PC cells were not associated with any corresponding increase in the respective mRNA levels (Figure 1F). Interestingly, MID1 can associate with microtubules to form large microtubule-bound multiprotein complexes with AR mRNA to increase AR mRNA translation independent of mRNA levels [13]. Its up-regulation by both Enz and Abi treatment in LNCaP cells (Figure 1F left panel) suggests that at least in these cells, MID1 may modulate AR in response to ARAT agents.

Some basal cleaved PARP-1 can be detected in both untreated LNCaP and VCaP cells (Figure 1G and Appendix A). Although both Enz and Abi enhance PARP-1 cleavage in VCaP cells, such a response is not observed in the LNCaP cells with ARAT treatment (Figure 1G). PARP-1 cleavage usually signals apoptotic programmed cell death (PCD), which is generally associated with annexin V binding to membrane phosphatidylserine residues. However, annexin V/propidium iodine (PI) staining demonstrated that Enz or Abi treatment did not appear to statistically significantly increase the percentage of annexin V^+^ apoptotic cells in LNCaP or VCaP cells beyond the background rate observed in the control cells (Table 1). Taken together, the above data demonstrate that the ARAT agents can increase AR or ARv7 expression and primarily have an anti-proliferative effect in LNCaP cells, while they can induce non-apoptotic PCD (i.e., no annexin V-staining) in VCaP cells associated with cleaved PARP-1.

### 2.2. Metformin Inhibits AR and ARv7 Expression and Induces Cell Death in PC Cells

We evaluated the effects of metformin on LNCaP and VCaP cells (Figure 2). Metformin inhibited LNCaP and VCaP cell proliferation as determined by the BrdU (Figure 2A) and MTT assays (Figure 2B). Inhibition was first observed at the 1 mM metformin dose, which resulted in approximately 20–30% inhibition in both cell lines. At metformin concentrations >1 mM, inhibition of PC cell proliferation occurred in a dose dependent manner (Figure 2A,B). Although the relative proportions of dead cells were small, consistently 1 mM metformin treatment for 5 days increased the absolute number of detached floating trypan blue positive cells compared to DMSO-treated control cells (from near 0% to 0.7% among LNCaP cells (Figure 2C left panel) and from 1.1% to 2.5% among VCaP cells (Figure 2D left panel)). Consistent with the BrdU and MTT assays, 1 mM metformin also significantly decreased the total attached cell number (mostly trypan blue negative) among LNCaP cells (Figure 2C right panel) and VCaP cells (Figure 2D right panel) compared to controls. The effects of metformin (1 mM × 5 days) on LNCaP and VCaP cells were also evaluated by flow cytometry after staining the cells for annexin V. As shown in Table 1, a statistically significant induction in apoptotic-like PCD (i.e., annexin V^+^) occurred in LNCaP but not VCaP cells with metformin treatment. This underscores the relevance of cellular context in terms of the type of PCD that is recruited (apoptotic-like vs. non-apoptotic) by different PC cell types.

Figure 2E,F and Appendix A show Western blots showing the effects of metformin (concentration course and time course) on AR and ARv7 expression in LNCaP and VCaP cells, respectively. These data show that metformin decreases AR (LNCaP and VCaP cells) and ARv7 (VCaP cells) protein levels at concentrations of 1 mM or less in the two PC cell lines. Further, a statistically significant decrease in AR and ARv7 mRNA also occured with metformin (Figure 2G). The effects of metformin on PARP-1 cleavage in LNCaP and VCaP cells as a function of dose and time are shown in Figure 2H and Appendix A; metformin induced PARP-1 cleavage in both cells. Others have shown that metformin can induce apoptotic cell death in cancer cells but at significantly higher concentrations (10–30 mM range) [14,33]. Our data show that metformin can have anti-proliferative effects, induce cell death, and begin to induce PARP-1 cleavage at concentrations ranging between 0.5–1 mM.

### 2.3. Metformin and ARAT Agents (Enz, Abi) Enhance Cell Death in PC Cells

Given the long doubling times of the LNCaP and VCaP cell populations (42 to 51 h; Appendix A) [34], the effects of the agents of interest (Abi, Enz, metformin, and ARAT + metformin combinations) on cell proliferation become apparent only after the longer incubation times. For instance, compared to controls, the anti-proliferative effects of the above treatments on LNCaP cells are observed at day 3 or beyond after the drug treatments (Appendix A). Trypan blue exclusion also did not show any cell death effects at earlier time points. Thus, in subsequent studies, we evaluated the combination treatments after 5 days of drug incubation. LNCaP (Figure 3A) and VCaP (Figure 3B) cells were evaluated with trypan blue staining after treating them with metformin without or with Enz or Abi. In Figure 3A,B left panels, floating dead cells as a percentage of total cells (floating + attached cells) are shown. Among the treated LNCaP cells the percentage of dead cells increased from approximately 0.7% with metformin alone to 4.8–5% with the combination treatments, and among the treated VCaP cells, the percentage of dead cells increased from about 2.5% with metformin alone to 12–15% with the combination treatments. Greater inhibition of cell proliferation is also noted with the combination of metformin plus ARATs compared to ARAT agents alone (Figure 3A,B, right panels). Additional studies in LNCaP cells using the combination index (CI) method of Chou and Talalay [35] demonstrate that metformin and ARATs together had an additive to synergistic anti-proliferative effect (Appendix A). In keeping with these observations, cell cycle analysis reveals that a greater proportion of the LNCaP cells were transitioned into the G0/G1 phase with the drug treatments, including ARAT + metformin combinations (Appendix A). Metformin + ARATs also induced statistically significant inhibition of colony formation in longer term 2D clonogenic assays at the tested doses in both LNCaP and VCaP cells (metformin 1 mM, Abi 5 µM, and Enz 10 µM) (Appendix A).

The effects of the different treatments on AR and ARv7 expression in LNCaP and VCaP cells are shown in Figure 3C–E and Appendix A. Although increase in expression of AR or ARv7 can occur upon treatment of LNCaP or VCaP cells with ARATs, metformin alone or in combination with Abi or Enz is associated with a decrease in AR/ARv7 protein (Figure 3C,D) and mRNA (Figure 3E) levels in both PC cells. Further, these effects become apparent in the combination treatments at relatively low concentrations of metformin (0.5–1 mM) (Appendix A).

Although the addition of ARATs to metformin increases cell death in both LNCaP and VCaP cells, only in the LNCaP cells was this associated with an increase in annexin V/PI staining (Table 1). Bcl-2/bax ratios appear to be decreased in LNCaP but not VCaP cells treated with metformin or metformin + ARATs (Figure 3F). Compared to treatment with the protein kinase inhibitor staurosporine that serves as a positive control for caspase 3-mediated cell death, we observed minimal/borderline caspase 3 cleavage in LNCaP cells and no caspase 3 cleavage in VCaP cells with metformin + ARATs (Figure 3F). Metformin-induced cleavage of PARP-1, on the other hand, was enhanced with the addition of ARATs in both cell lines (Figure 3G), and these effects decrease upon silencing of PARP-1 (Figure 3H). Thus, cPARP-1-associated PCD was either minimally (LNCaP) or essentially not (VCaP) associated with caspase 3 activation in PC cells treated with metformin + ARATs.

The combination treatments also led to an activation of PARP-1, as reflected by the increased formation of PAR in both LNCaP and VCaP cells (Figure 3G). Since PAR can induce nuclear translocation of cleaved AIF and result in PAR-AIF-mediated cell death, we explored the effect of the combination treatments on PAR production and the induction of AIF nuclear translocation. Nuclear extracts from LNCaP (Figure 3I left panel) and VCaP cells (Figure 3I right panel)) treated for 5 days with metformin, or the combinations were evaluated via immunoblotting using an anti-cAIF antibody, with histone H3 serving as a loading control. Consistent with the increased production of PAR, metformin + ARATs increased the accumulation of cAIF in the nucleus (Figure 3I). Interestingly, we also noted an increase in cAIF in the nucleus of LNCaP cells, and to a lesser extent in VCaP cells, with single agent metformin, but these effects appeared to occur independently of PARP-1 activation (Figure 3I,G).

### 2.4. Metformin with or without ARATs Increases Lysosome Permeability and Cathepsin G-Mediated PARP-1 Cleavage in Androgen-Sensitive PC Cells

Enhanced lysosomal membrane permeability (LMP) in response to certain cellular stresses can cause leakage of lysosomal proteases, particularly cathepsins, which in turn contribute to some of the early events associated with apoptotic or apoptotic-like PCD that include cleavage of certain cytosolic proteins such as PARP-1, among others [36,37]. As shown in Figure 4A (LNCaP cells) and Figure 4B (VCaP cells), the intense staining of acidic vacuoles (lysosomes) by acridine orange in control cells or cells treated with Enz or Abi became diffuse and dim when the cells were treated with metformin or metformin + ARATs, suggesting that metformin-based treatments decrease lysosome acidity, perhaps due to an increase in LMP. In keeping with this, inhibition of lysosome activity with 100 µM chloroquine abolished metformin induced PARP-1 cleavage in both LNCaP cells and VCaP cells (Figure 4C and Appendix A). The cysteine protease inhibitor, E-64d, which blocks the activity of cathepsins, also abolished metformin induced PARP-1 cleavage in both LNCaP cells and VCaP cells (Figure 4D). Furthermore, silencing the cysteine protease cathepsin G with siRNA abolished the metformin induced cleavage of PARP-1, (Figure 4E). However, interestingly such an effect was not observed in cathepsin D silenced cells (Figure 4F), although this lysosomal protease has been implicated in PCD in other model systems [38,39]. Finally, trypan blue staining of LNCaP cells demonstrated that metformin- or metformin + Enz-induced cell death is inhibited by either chloroquine or E-64d (Figure 4G), consistent with the role of lysosomes in metformin-mediated cell death in PC cells.

## 3. Discussion

Although hormonally directed therapies can result in clinically meaningful responses in patients with PC, such therapies are primarily palliative and work for only a limited period in most patients. A major challenge, especially with agents such as abiraterone and enzalutamide, is how to improve their anti-tumor activity while also maintaining treatment safety. Multi-faceted approaches that target not only the hormonal axis but other pathways are being actively evaluated to improve the anti-prostate cancer activity of such approaches.

In addition to its glucose lowering properties, metformin has effects on mitochondrial function and cellular metabolism, including mitigation of hyperinsulinemia and activation of the AMP-kinase pathway, which may contribute to some of its purported anti-proliferative and anti-tumor properties [40,41]. Insulin is a growth factor, and hyperinsulinemia present in insulin resistance is associated with lower levels of sex hormone binding globulin, thereby increasing the availability of unbound free androgens. Given that there is wide clinical experience and an established safety profile with metformin, it is a particularly attractive agent to repurpose into combination regimens for anti-cancer therapy, as is being done with some other previously approved drugs [42] In the present study, we employed two independent well-established cell culture models of human PC, i.e., LNCaP and VCaP cells, to further study and clarify the potential role of metformin in the context of targeting the androgen/AR axis under the backdrop of different molecular characteristics that define these two cell lines. The two cell lines retain relative sensitivity to androgens, and share some but not other underlying biological properties, thus representing some of the heterogeneity seen within the clinical PC disease spectrum. For instance, LNCaP cells express wt p53, mutant (but functional) AR, and mutant PTEN, whereas VCaP cells have abrogated p53 function (due to p53 allelic deletion and missense mutation), express both AR and ARv7, and have wt PTEN

By using several complementary assays, we demonstrate that ARATs, metformin, and ARATs + metformin lead to growth inhibition in both LNCaP and VCaP cells, with greater and statistically significant inhibitory effects noted with the combination treatments. The combination of metformin + ARATs consistently increased the percentage of TB^+^ (dead) cells compared to untreated- or single agent-treated cells in both cell types. In addition, the proportion of attached mostly TB^−^ (alive) cells was decreased further when metformin was added to ARATs compared to ARAT treatments alone.

Fluorescence-activated cell sorting (FACS) analysis (Table 1) demonstrated that metformin or metformin + ARAT treatment was associated with increased annexin V staining in LNCaP, but not VCaP cells. Annexin V binds to externalized phosphatidylserine on cell plasma membranes, which is one of the earliest events in several but not all forms of PCD. Interestingly, we also found differential annexin V staining between LNCaP and VCaP cells in response to the potent inducer of PCD, staurosporine; enhanced caspase 3 cleavage occurred in both cell types, but an increase in annexin V staining was observed only in LNCaP cells (Table 1). Taken together, these data suggest that VCaP cells may recruit a cell death program that shares many but not all features of a canonical PCD pathway typically associated with annexin V staining.

We found ARATs enhanced cleavage of PARP-1 in VCaP but not LNCaP cells (Figure 1G). However, metformin (1 µM) as a single agent induced PARP-1 cleavage in both cell lines within 4–5 days of treatment (Figure 2H). Metformin in combination with ARATs is particularly effective in inducing cleavage of PARP-1 (Figure 3G), which is further underscored by the demonstration of PARP-1 cleavage with the combination even after PARP-1 knockdown with siRNA (Figure 3H). The combination treatments also enhanced PARP-1 activity, as evidenced by a dramatic increase in PAR levels in the ARAT + metformin treated cells (Figure 3G). Increased PAR production results in its translocation from the nucleus to the mitochondria causing a release of AIF (cleaved form) from the latter. The cleaved AIF, in turn, recruits DNA endonuclease to the nucleus, which leads to DNA cleavage and a form of PCD termed parthanatos [26,43]. Consistent with this paradigm, ARATs + metformin led to increased cAIF in the nuclear fractions of both LNCaP and VCaP cells. A putative mechanism of cell death by cleaved PARP-1 is that it binds to DNA, preventing non-cleaved PARP-1 from accessing the damaged sites and initiating repairs [44]. Therefore, a relatively small amount of cleaved PARP-1, as observed in our studies, may be enough to block further DNA repair, while the remaining non-cleaved PARP-1 can produce more PAR and thus also contribute to PCD (Figure 3G). In other studies, we evaluated the potential role of lysosomal proteases in mediating some of the effects of metformin and ARATs in the PC cells since they can amplify the cell death program. Indeed, metformin or metformin, in combination with ARATs, but not ARATs alone, increased lysosomal membrane permeability (LMP) and PARP cleavage in the PC cells, with a partial abrogation of cell death occurring when LMP was inhibited.

ADT and ARATs have proven to be among the most effective anti-PC agents to date clinically, particularly in the castration-sensitive setting. Although this degree of clinical efficacy of ADT/ARATs is perhaps not as adequately reflected in in vitro PC models in that ADT and ARATs primarily induce an anti-proliferative response in cell culture models of androgen responsive PC cells, nevertheless such models provide useful pre-clinical signals that can inform anti-cancer agent activity in the clinical setting. In this regard, our data show that metformin, when added to ARATs, can enhance the anti-proliferative activity of ARATs and increase PCD, and thus are of potential clinical relevance. However, some of the limitations of our study are that it was limited to two androgen-responsive PC cell lines and did not evaluate the combinations in either androgen-insensitive or AR negative PC cells, and was also restricted to in vitro models. Further, the study did not evaluate some of the other effects the combination treatments could have had on the metabolome. The question also remains as to whether adequate levels of metformin to effect anti-tumor responses can be achieved in patients. Some of the failed clinical trials with metformin in non-diabetic cancer patients to date may be due to the lower doses of metformin used in the clinic as compared to pre-clinical studies. During controlled clinical trials, the maximum dose of metformin hydrochloride tablets did not exceed 2550 mg daily, resulting in maximum metformin plasma levels of less than 5 µg/mL (about 30 µM). We tested the effects of metformin at significantly lower doses (1 mM or less) than have been reported by many other investigators in pre-clinical studies (generally 5–30 mM). Although achieving such levels of metformin in plasma remains a challenge, it is transported into cells by the organic cation transporter 1-3 (OCT1-3), which can be highly expressed in prostate tissue and may allow for enhanced intracellular drug accumulation [45,46].

In conclusion, we demonstrate that metformin in combination with ARATs causes enhanced anti-proliferative effects and also induces cell death via pathways other than the canonical apoptotic machinery. The combination results in the recruitment of two PARP-1-dependent cell death pathways, including via enhanced cleavage of PARP-1 and enhanced production of PAR with an associated increase in nuclear cAIF accumulation (Figure 5). Metformin/ARAT-mediated parthanatos, to our knowledge, has not been described previously in PC. Our study adds to the growing body of evidence regarding the potential range of mechanisms that can mediate anti-tumor effects of metformin in concert with ARATs, including, for instance, the recent demonstration of metformin sensitizing PC cells to enzalutamide via recruitment of STAT3/TGFb signaling [47]. Many of the initial trials with metformin in PC have evaluated it in the advanced castration resistant setting, a disease state that is generally more refractory to additional therapies compared to castration sensitive disease. Given that metformin enhances the anti-cellular effects of ARATs in androgen-responsive cells, it will be of interest to evaluate prospectively metformin/ARAT-based combinations, particularly in treatment-naïve, castration-sensitive settings of PC. Indeed, large retrospective analysis has demonstrated that diabetic PC patients initiated on ADT who are on metformin have statistically significant better overall and cancer specific survival compared to diabetic PC patients on ADT but not metformin [48]. Finally, the present study also provides a framework to test whether anti-tumor efficacy can be further improved in PC by incorporating other rationally selected targets into a metformin/ARAT backbone.

## 4. Materials and Methods

### 4.1. Materials and Drugs

Acridine orange, metformin, 3-(4,5-dimethylthiazol-2-yl)-2,5-diphenyltetrazolium bromide (MTT), chloroquine, and E-64d were purchased from Sigma-Aldrich, Inc. (St. Louis, MO, USA). Anti-AIF, anti-PAR, and anti-cathepsin G antibodies were purchased from Santa Cruz BioTechnology (Santa Cruz, CA, USA). Antibodies to PARP-1, cleaved PARP-1, Bcl2, Bax, Mcl1, caspase 3, cleaved caspase 3, AR, ARv7, actin and GAPDH, and the horseradish peroxidase labeled secondary antibody were purchased from Cell Signaling Technology (Danvers, MA, USA). All antibodies were used at 1 to 1000 dilution. Enzalutamide (Enz) was purchased from Selleck Chemicals (Houston, TX, USA). Abiraterone acetate (Abi) was a gift from Johnson and Johnson Health Care Systems Inc (New Brunswick, NJ, USA).

### 4.2. Cell Lines and Culture Conditions

VCaP cells (obtained from ATCC, used at passage numbers less than 40) were grown in DMEM/F12 (1:1) media (BioSource, Grand Island, NY, USA) supplemented with 5% FBS (Biosource), 50 units/mL penicillin, and 50 µg/mL streptomycin. LNCaP cells (obtained from ATCC, used at passage numbers less than 30) were cultured in RPMI1640 medium (BioSource, Grand Island, NY, USA) supplemented with 10% FBS (Biosource), 50 units/mL penicillin, and 50 µg/mL streptomycin. The culture medium was changed every other day. For all cell culture experiments, the cell passage number of wild-type LNCaP cells was 34 or less while that for VCaP cells was 50 or less.

### 4.3. RNA Preparation and Quantitative RT-PCR (qPCR)

Wild-type cells were plated at a density of 3 × 10^5^ cells/per well in 35 mm petri dishes. The cells were treated singly with metformin (1 mM), Enz (10 µM), or Abi (5 µM), or in specific combinations for 5 days. Total mRNA was extracted using Trizol (Life Technology, Carlsbad, CA, USA) and isolated with RNA mini-prep columns (Qiagen, Valencia, CA). The mRNA was reverse transcribed to cDNA with M-MLV reverse transcriptase (Roche Diagnostics, Basel, Switzerland). Reverse-transcribed cDNA (10–100 ng) was amplified by real-time PCR using IQ SYBR green mix (Bio-Rad, Hercules, CA, USA) and detected via the MyiQ Single-Color Real-Time PCR Detection System (Bio-Rad). Each reaction was performed in duplicate. The sequences of AR, Arv7, MID1, and TBP primers are provided in Table 2. The sequences of Arv7 primers used are in reference [49].

### 4.4. Preparation of Protein Extracts and Western Blot Analysis

We cultured 3 × 10^5^ cells/mL in 35 mm plates. Following the indicated treatments, cells were washed in 1 × PBS; the attached cells were lysed in radioimmune precipitation buffer for 30 min on ice with occasional vortexing. The clarified lysates were separated by 4–12% SDS-polyacrylamide gel electrophoresis and analyzed by Western blotting using the relevant primary antibodies as indicated. The bands were visualized by enhanced chemiluminescence (GE Healthcare Bio-Sciences Corp., Piscataway, NJ, USA) and quantified by densitometric analysis (Visionworks LS image acquisition and analysis software, UVP, Upland, CA, USA).

### 4.5. Isolation of Nuclear Fractions

PC cells were seeded in 60 mm dishes and treated with various drugs for 5 days. Nuclear extracts from the cells were prepared using the NE-PER extraction kit (Pierce Thermo Scientific Inc., Rockford, IL, USA) and protein quantified using the BCA assay kit (Pierce Thermo Scientific Inc., Grand Island, NY, USA).

### 4.6. MTT Assay

MTT assay was performed essentially as described by Mossman [50]. LNCaP cells were plated in 96-well plates at a density of 3 × 10^3^ cells/well in 100 µL culture medium supplemented with 10% FBS. After two days, cells were treated with DMSO or the agents under study. After the treatment period, 25 µL/well of 5 mg/mL MTT stock solution was added for 3 h, the media subsequently removed, and the resulting formazan crystals dissolved in isopropanol (200 µL/mL) and optical density (OD) determined at 570 nm.

### 4.7. Trypan Blue Staining Assay

Cell viability was assessed via trypan blue staining. PC cells were plated in 6-well plates at a density of 1 × 10^6^ cells/well. Drugs were added to the plates (in duplicates), and after 5 days of incubation, both floating and attached cells were collected separately, stained with trypan blue, and counted using a hemocytometer.

### 4.8. Annexin V Detection

Annexin V was detected using the FITC Annexin V Apoptosis Detection kit from BD Biosciences. PC cells were treated with various drugs, washed in 1 × PBS, and the attached cells harvested and incubated with Annexin V/PI according to the manufacturer’s instructions, with cell staining subsequently assessed by flow cytometry.

### 4.9. Lysosome Staining by Acridine Orange

Cells grown on coverslips and treated with study drugs singly or in various combinations for 5 days were washed with 1 × PBS and then stained with 1 mM acridine orange (Sigma A6014) at 37 °C for 15 min to label acidic lysosomes. Excess acridine orange was washed with 1 × PBS and the cells were examined via a Nikon Eclipse 80i (Nikon Instruments Inc., Melville, NY, USA) wide field fluorescent microscope.

### 4.10. siRNA Interference

Knock down of cathepsin D, cathepsin G, and PARP-1 mRNA was performed using pre-designed cathepsin D or G siRNA (final concentration 10 nM, Santa Cruz, Dallas, TX, USA), or PARP-1 siRNA (final concentration 10 nM, Cell Signaling Technology). Non-specific siRNA (10 nM, Santa Cruz, Santa Cruz, Dallas, TX, USA) was used as a negative control. Cells were seeded in 35 mm Petri dishes and transfected with siRNA using Lipofectamine RNAiMAX (Invitrogen, Carlsbad, CA, USA) according to the manufacturer’s protocol, then treated with the relevant drugs as indicated 24 h post transfection and incubated for 3 days before cell harvest.

### 4.11. Drug Combination Index

Drug combination studies were performed according to the methods described by Chou and Talalay [35]. Three thousand cells/well were seeded in 96-well plates and allowed to attach over 48 h. Drugs were then added at their fixed IC50 ratios at various concentrations as a single agent or in combination, and cells were incubated for 3 days. MTT assays were carried out as described above. The combination index (CI) values at 50%, 75%, and 90% of effective doses and dose reduction index (DRI) values for each drug in the combination were determined using the CalcuSyn 2.1 program [51].

### 4.12. 2-D Clonogenic Assays

We plated 1 × 10^4^ cells per well in 12-well plates. After 48 h, cells were treated with DMSO (control), Abi, Enz, metformin, or combinations and incubated for 10 days. After incubation, media was aspirated, colonies washed with PBS, and fixed with 200-proof ethanol for 30 min. Colonies were then stained with 0.5% crystal violet for 30 min at room temperature, and extra stain washed. The surface area of the wells covered by the colonies was assessed using ImageJ 1.48v software (NIH, Bethesda, MD, USA) (LNCaP cells). CellCounter (https://nghiaho.com/ (accessed on 28 February 2020)) software by Nghia Ho [52] was used to determine the number of colonies and graphed with GraphPad Prism (v 8.3.1) (VCaP cells). The standard error of mean (SEM) was generated for each treatment from three different wells. A Student’s *t*-test was performed to determine *p*-values. Three independent experiments were conducted for biological validation of the data. 

### 4.13. Statistical Methods

Student’s *t*-tests were used for statistical comparisons.

## 5. Conclusions

We demonstrate that metformin in combination with abiraterone or enzalutamide has prominent anti-proliferative effects, decreases AR and ARv7 levels, and induces cell death in androgen sensitive prostate cancer cells via recruitment of two PARP-1-dependent cell death pathways, including via enhanced cleavage of PARP-1 and enhanced production of PAR with an associated increase in nuclear cAIF accumulation.

## Figures and Tables

**Figure 1 cancers-13-00633-f001:**
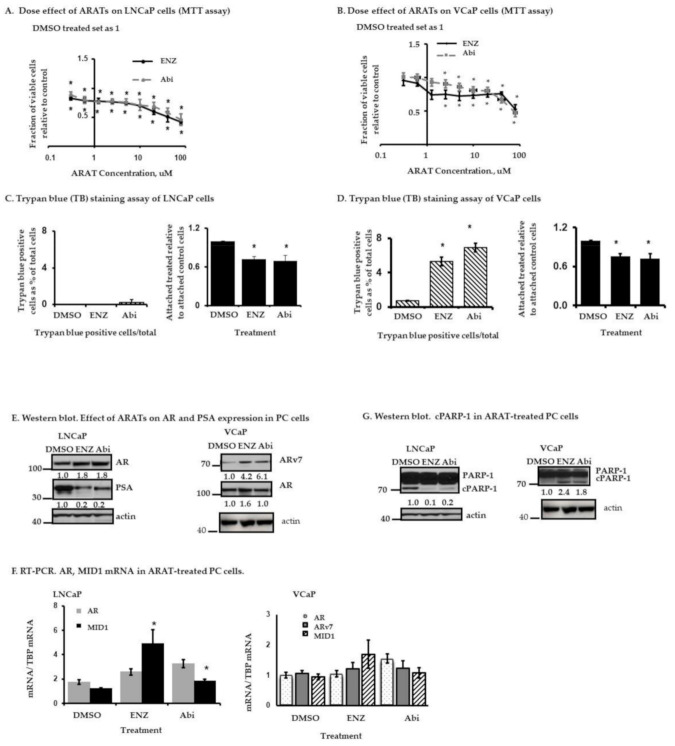
Effect of enzalutamide (Enz) and abiraterone (Abi) on LNCaP and VCaP cells. (**A**,**B**) 3-(4,5-dimethylthiazol-2-yl)-2,5-diphenyltetrazolium bromide (MTT) assay. Cells were treated with increasing doses of Enz or Abi for 3 days. Shown are the fractions of viable treated cells relative to control (DMSO-treated) cells, as determined by MTT assay. Results represent the mean ± standard deviation (S.D.) from three separate experiments, each done with triplicate determinations per data point. (* *p* < 0.05), Student’s *t*-test. (**C**,**D**) Trypan blue staining. Cells were treated with Enz 10 µM, Abi 5 µM or DMSO (control) for 5 days and stained with trypan blue. Both floating and attached cells were collected. Floating cells generally represent dead cells (uniformly stained with trypan blue), while attached cells are not yet dead (exclude trypan blue). (**C**) Left. LNCaP floating cells are shown as a percentage of total treated cells (floating + attached) within each plate. Right. Drug-treated LNCaP attached cells are shown as a fraction of attached control (DMSO-treated) cells, which was set at 1. (**D**) Left. VCaP floating cells are shown as a percentage of total treated cells (floating + attached) within each plate. Right. Drug-treated VCaP attached cells as a fraction of attached control (DMSO-treated) cells, which was set at 1. Results represent the mean ± S.D. from three separate experiments, each done with triplicate determinations per data point. (* *p* < 0.05), Student’s *t*-test. (**E**) Western blot. Androgen receptor (AR), ARv7, and prostate serum antigen (PSA) expression in PC cells treated with Enz 10 µM or Abi 5 µM for 5 days. (**F**) RT-PCR. Relative AR, ARv7, and MID1 mRNA expression in 5 day-treated PC cells. The specific mRNAs were normalized with the respective TATA-box-binding protein (TBP) mRNA in the drug-treated or DMSO (control)-treated cells. Results represent the mean ± S.D. from three separate experiments, each done with duplicate determinations per data point. (* *p* < 0.05), Student’s *t*-test. (**G**) Western blot. PARP-1 and cPARP-1 expression in PC cells treated for 5 days with Enz or Abi. Results are representative of two separate experiments. The original western blot figures are available in a separate Appendix A document. In (**E**) and (**G**), the actin loading controls for the respective LNCaP and VCaP western blots are identical since they are part of the same original blots probed with different antibodies of interest; the figures are organized as shown to better align with the data presented in the Results Section.

**Figure 2 cancers-13-00633-f002:**
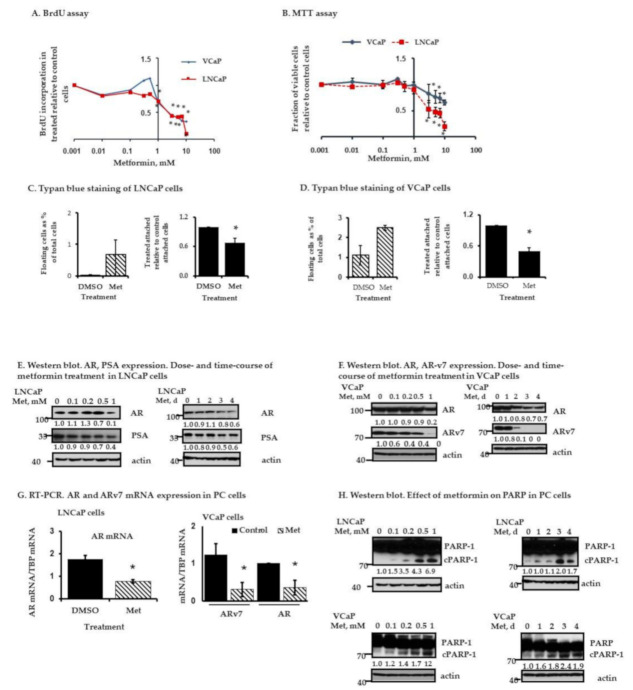
Effect of metformin on LNCaP and VCaP cells. (**A**) BrdU assay. Effect of metformin dose on cell proliferation. Shown is BrdU incorporation in 3 days metformin-treated cells as a fraction of control (DMSO-treated) cells. Results represent the mean ± S.D. from three separate experiments, each done with triplicate determinations per data point. (* *p* < 0.05), Student’s *t*-test. (**B**) MTT assay. Cells were treated with increasing doses of metformin for 3 days. Shown are the fractions of viable treated cells relative to control (DMSO-treated) cells. Results represent the mean ± S.D. from three separate experiments, each done with triplicate determinations per data point. (* *p* < 0.05), student’s *t*-test. (**C**,**D**) Trypan blue staining. Cells were treated with metformin 1 µM or DMSO (control) for 5 days and stained with trypan blue. Both floating and attached cells were collected. (**C**) Left. LNCaP floating cells shown as a percentage of total treated cells (floating + attached) within each plate. Right. Metformin-treated LNCaP attached cells are shown as a fraction of attached control (DMSO-treated) cells (set as 1). (**D**) Left. VCaP floating cells shown as a percentage of total treated cells (floating + attached) within each plate. Right. Metformin-treated VCaP attached cells are shown as a fraction of attached control (DMSO-treated) cells (set as 1). Results represent the mean ± S.D. from two separate experiments, each done with triplicate determinations per data point. (* *p* < 0.05), Student’s *t*-test. (**E**) Western blot. AR, PSA expression in metformin-treated LNCaP cells. Left, dose course (cells were treated with increasing concentrations of metformin for 4 days). Right, time course (cells were treated with metformin 1 mM for a different number of days). (**F**) Western blot. Same as in (**E**), metformin treated dose-course and time-course in VCaP cells. Westerns are representative of two independent blots. (**G**) RT-PCR. Relative AR (LNCaP cells) or AR, ARv7 (VCaP cells) mRNA expression in metformin-treated (1 µM × 5 days) or DMSO (control)-treated cells normalized to TBP mRNA in the respective cells. Results represent the mean ± S.D. from three separate experiments, each done with duplicate determinations per data point. (* *p* < 0.05), Student’s *t*-test. (**H**) Western blot. Effect of metformin-treated dose course and time course on PARP-1 cleavage in LNCaP and VCaP cells. Results are representative of two independent blots. The original western blot figures are available in a separate Appendix A document. In (**E**) and (**H**) the actin loading controls for the respective LNCaP western blots are identical, and in (**F**) and (**H**) the actin loading controls for the respective VCaP western blots are identical since they are part of the same original blots probed with different antibodies of interest; the figures are organized as shown to better align with the data presented in the Results Section.

**Figure 3 cancers-13-00633-f003:**
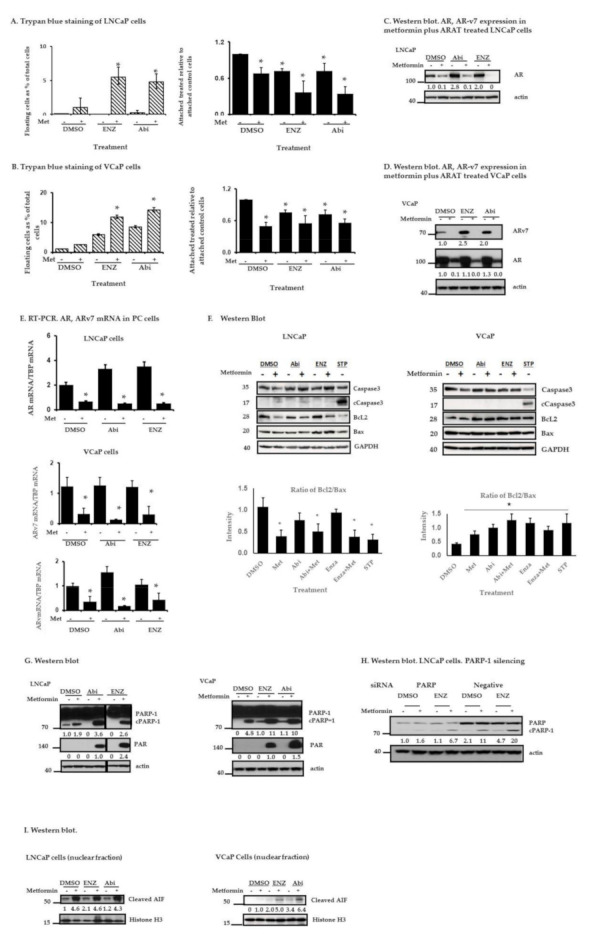
Treatment of PC cells with metformin in combination with ARAT agents (Enz, and Abi). (**A**,**B**) Trypan blue staining. Cells were treated for 5 days with metformin 1 µM or Enz (10 µM) ± metformin or Abi (5 µM) ± metformin, then stained with trypan blue. Both floating and attached cells were collected. (**A**) Left. LNCaP floating cells shown as a percentage of total treated cells (floating + attached) in each plate. Right. Treated LNCaP attached cells are shown as a fraction of attached control (DMSO-treated) cells (set as 1) (**B**) Left. VCaP floating cells shown as a percentage of total treated cells (floating + attached) in each plate. Right. Treated VCaP attached cells are shown as a fraction of attached control (DMSO-treated) cells (set as 1). Results represent the mean ± S.D. from two separate experiments, each done with triplicate determinations per data point. (* *p* < 0.05), Student’s *t*-test. (**C**,**D**) Western blot. AR, ARv7 expression in PC cells treated for 5 days with metformin, Enz ± metformin, or Abi ± metformin. The Western blots shown are representative of two independent experiments. (**E**) RT-PCR. Relative AR (LNCaP cells) or AR, ARv7 (VCaP cells) mRNA expression in metformin-treated, Enz ± metformin-treated, Abi ± metformin-treated or control (DMSO-treated) cells normalized to TBP mRNA in the respective cells. Results represent the mean ± S.D. from three separate experiments, each done with duplicate determinations per data point. (* *p* < 0.05), Student’s *t*-test. (**F**) Western blot. Caspase 3, Bcl2, Bax in PC cells treated with metformin (1 mM), Enz (10 µM) ± metformin, or Abi (5 µM) ± metformin for 5 days, or staurosporine 1 µM for 5 hrs. Results are representative of two independent experiments. (**G**) Western blot. cPARP-1 and PAR in PC cells treated with metformin, Enz or Abi, as in (**F**). (**H**) Western blot. Effect of PARP-1 silencing on PARP-1 cleavage in LNCaP cells treated with metformin (1 mM), Enz (10 µM), or both for 5 days. (**I**) Western blot. Cleaved apoptosis inducing factor (AIF) in LNCaP and VCaP cells treated with metformin (1 mM), Enz (10 µM) ± metformin, or Abi (5 µM) ± metformin for 5 days. Histone H3 served as a nuclear loading control. The original western blot figures are available in a separate Appendix A document.

**Figure 4 cancers-13-00633-f004:**
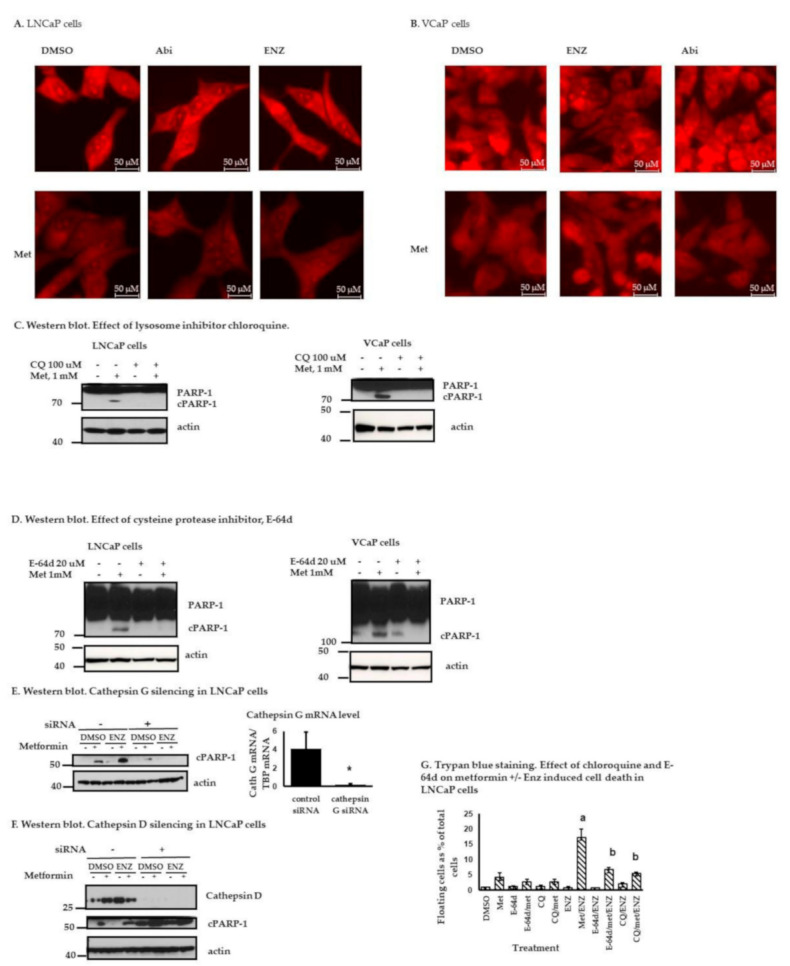
Lysosome- and cathepsin G-mediated effects in PC cells. (**A**,**B**) Acridine orange staining of LNCaP cells (**A**) and VCaP cells (**B**) after treatment with metformin (1 mM), Enz (10 µM) ± metformin, or Abi (5 µM) ± metformin for 5 days demonstrate enhanced lysosome permeability with metformin or combination treatments. (**C**,**D**) Western blot. Effect of lysosome inhibition by chloroquine (100 µM) (**C**) or E-64d (**D**) on metformin (1 mM for 5 days)-mediated PARP-1 cleavage in LNCaP or VCaP cells. (**E**,**F**) Effect of silencing cathepsin G (**E**) or cathepsin D (**F**) by siRNA on PARP-1 cleavage in LNCaP cells treated with metformin 1 mM ± Enz 10 µM for 5 days. The original western blot figures are available in a separate Appendix A document. (**G**) Trypan blue staining. LNCaP cells were treated for 5 days with inhibitors (chloroquine, E-64d) or drug (metformin, Enz) or both as shown, stained with trypan blue, and floating and attached cells collected. Shown are floating cells as a percentage of the total (floating + attached) treated cells. Results represent the mean ± S.D. from three separate experiments, each done with triplicate determinations per data point. a, (* *p* < 0.05) comparing with DMSO treated group, b, (* *p* < 0.05) comparing with Enz/metformin treated group, Student’s *t*-test.

**Figure 5 cancers-13-00633-f005:**
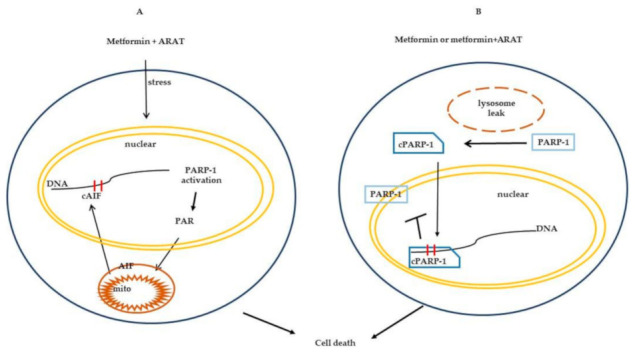
Schematic: recruitment of PARP-1-dependent cell death in PC cells by metformin and ARAT (Abi, Enz) combinations. (**A**) Cellular stress induced by metformin/ARATs leads to PARP-1 activation, increase PAR production and cAIF nuclear accumulation. (**B**) Metformin or metformin/ARATs induce cathepsin leak from lysosomes resulting in cleavage of PARP-1, cPARP-1 then binds DNA to prevent uncleaved PARP-1 from accessing DNA for repair.

**Table 1 cancers-13-00633-t001:** Annexin V/PI staining of LNCaP and VCaP cells treated with metformin, ENZ, Abi, or staurosporine as determined by flow cytometry.

Cells	DMSO	Metformin	ENZ	ENZ/met	Abi	Abi/met	STP
LNCaP cells	9.6 ± 2.8	16.6 ± 3.3 *	19.8 ± 9.8	27.2 ± 6.8 *	16.3 ± 6.0	25.8 ± 4.85 *	17.5 ± 2.7 *
VCaP cells	3.7 ± 2.3	4.8 ± 3.2	4.5 ± 3.4	6.0 ± 4.2	4.7 ± 3.8	5.0 ± 3.4	5.7 ± 0.3 *

Annexin V/propidium iodide (PI) staining of prostate cancer (PC) cells after drug treatment with metformin (met), enzalutamide (ENZ), abiraterone (Abi), or combinations for 5 days. Cells were treated with straurosporine (STP) 1 µM for 5 h. Results represent the mean ± S.D. from three separate experiments; * (*p* < 0.05) vs. DMSO treated, Student’s *t*-test.

**Table 2 cancers-13-00633-t002:** Primer pairs used for qRT-PCR.

AR_Fw	gctctacttcgcccctgatc
AR_Rv	ttcggacacactggctgtac
MID1_Fw	ctgggtcagccattttgact
MID1_Rv	tatttcagggaggcagttgg
TBP_Fw	tgcccgaaacgccgaatata
TBP_Rv	cgtggttcgtggctctctta

## Data Availability

Data sharing not applicable.

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
