# Peer review of "Metformin and Androgen Receptor-Axis-Targeted (ARAT) Agents Induce Two PARP-1-Dependent Cell Death Pathways in Androgen-Sensitive Human Prostate Cancer Cells"

_cancers, 2021, doi:10.3390/cancers13040633_

Round 1

Reviewer 1 Report

The authors presented data on Metformin and Abiraterone or Enzalutamide combination for the Androgen-sensitive PC cells LNCaP and VCaP. The manuscript is much improved compared to the previous version and addressed almost all the concerns. Some minor points are:

  1. The quality of the figures need to be improved, especially the legend size within the figures; different fonts.
  2. The position of Table 1 should be at a later page, closer to the text description.
  3. Figure S1A and B are the same thing, only one is needed.
  4. Some statistical analysis are missing in S3B.    

Author Response

Comments and Suggestions for Authors from Reviewer 1

The authors presented data on Metformin and Abiraterone or Enzalutamide combination for the Androgen-sensitive PC cells LNCaP and VCaP. The manuscript is much improved compared to the previous version and addressed almost all the concerns. Some minor points are:

  1. The quality of the figures need to be improved, especially the legend size within the figures; different fonts.
  2. The position of Table 1 should be at a later page, closer to the text description.
  3. Figure S1A and B are the same thing, only one is needed.
  4. Some statistical analysis are missing in S3B.  

Response to reviewer 1:

  1. We have changed the legends within the figures to be consistent in terms of font and size.
  2. Table 1 has been moved close to the text description as suggested.
  3. Figure S1A has been removed.
  4. Statistical analysis has been added in S3B.

Reviewer 2 Report

Though the authors have done most of the experiments as suggested in round 1, inconsistencies still remain. Western blots still have Actin in some places and GAPDH in some places as loading control. 

Author Response

Comments and Suggestions for Authors from Reviewer 2

Though the authors have done most of the experiments as suggested in round 1, inconsistencies still remain. Western blots still have Actin in some places and GAPDH in some places as loading control.

Response to reviewer 2:

We apologize for not correcting the inconsistencies in loading control. We have gone back and re-blotted the old membranes with actin to replace GAPDH in Figures 1E, 1G and Figures 4C, 4D as requested, but not for Figure 3F because we inadvertently did not save this specific membrane. However, we note that GAPDH is used as loading control for both the LNCaP and VCaP cells for Figure 3F and hence the loading control is consistent within the Figure.  

Round 2

Reviewer 2 Report

The manuscript has been appropriately revised. I think it is acceptable now.

This manuscript is a resubmission of an earlier submission. The following is a list of the peer review reports and author responses from that submission.

Round 1

Reviewer 1 Report

The authors presented data on Metformin and Abiraterone or Enzalutamide combination for the Androgen-sensitive PC cells LNCaP and VCaP. The manuscript is poorly prepared. 

1. Some figures are arranged in odd order (e.g. EFG before ABCD); tiny letters, different fonts, missing parts and cut off of western bands in Figures.
2. Most references are from 2015 or earlier, with only 3 newer ones. The authors did not mention the results from the clinical studies on Met/Abi and Met/ENZ on PC.
3. Discussion should not be the rewrite of the Results.
4. The study was carried out in Androgen-sensitive PC cells, yet the authors’ conclusion are for PC in general in many places.
5. How did the authors choose the 5-day treatment? To conclude that “neither Enz nor Abi induced cell death”(Line107) by evaluating only for day 5 is not accurate. Evaluation for each day should be carried out. Same thing goes for Met treatment and combination treatments.
6. IC50 value presented for Abi and Enz did not match the data in figures.
7. Any cell cycle data from the PI staining?

Reviewer 2 Report

General comments and abstract

  • The manuscript could benefit from editing for grammar, missing words, and subject-verb agreement, etc. It is recommended that authors delete irrelevant "general" phrases and sentences, repeated and unneeded words. They should use short sentences. Also, some Introductory sentences are irrelevant or are not needed. It is also recommended that authors send their manuscript to an expert in English editing and academic writing.
  • The abstract could be improved. The flow of ideas is not well organized. Authors started right away with the aim without giving a background on the topic to be studied.
  • Methodology lacks from the abstract. Authors jumped directly from the aim to the results.
  • The abstract lacks a solid conclusive statement. What is the clinical correlate of the study and results?
  • All abbreviations should be revised and defined at their first use.
  • For all Western blot figures, please include densitometry readings/intensity ratio of each band. In addition, please include the whole blot (uncropped blots) showing all the bands with all molecular weight markers on the Western in the Supplemental Materials

Introduction

  • Some references that the authors have used might not be up to date. Authors are kindly asked to review the whole manuscript and check the references accordingly.
  • First statement lacks a reference. I suggest citing Siegel et al. 2020.
  • In the second paragraph, authors elaborated on Metformin role in PC. It would be interesting to discuss its role in other tumors in the context of drug repurposing and cite up-to-date papers such as:doi: 10.1007/s10555-019-09840-2.
  • Elaborate more on ADT and its role in PC progression

Methodology

  • Methods section is weak and should be revised. More details should be added.
  • There are very few places where the details are missing in the study, which if added, might help the readers to reproduce the experiments.
  • The methodology part lacks elaboration in the statistical analysis section.
  • Add the dilution ratios for each antibody used.
  • Justify the doses of drugs used by citing references.

Results

  • Figure 1 format needs fixing. Start with sections A then B and so on instead of presenting section E first.
  • Add scale bars where appropriate (Fig 4 A and B)

Discussion

  • The discussion section is well written.
  • Authors elaborated and discussed their results in the context of what has been previously found in literature in a proper way.
  • Authors should have added a limitations section at the end of the discussion as their study has many limitations.

Reviewer 3 Report

Metformin and Androgen Receptor-Axis -Targeted 3 (ARAT) Agents Induce Two PARP-1-dependent Cell Death Pathways in Androgen-sensitive Human Prostate Cancer Cells is an attempt to enhance effectiveness of AART agents using routinely used and generally safe anti-diabetic agent Metformin. However, the study is lacking in many aspects and is not appropriate for Cancers.

  1. The authors treated prostate cancer cell lines with ARAT agents for 3 days before doing MTT assay. It is not appropriate. Being dependent on active cellular metabolism, MTT is reliable to determine viability at lower time points eg 12 to 24 hours. 72 hours is too long.
  2. Again, they emphasize a lot on trypan blue dye exclusion assay which is not very reliable. Alternatively, cell cycle analysis could be done.
  3. What is the doubling tie of these cells used? Treating them for 5 days with Enz and Abi to check gene expression is not acceptable. They need to check expression starting at least at 12h.
  4. Similarly, with metformin, treatment has been carried out for 5 days and too much reliance has been placed on trypan blue dye exclusion assay.
  5. PARP silencing using siRNA is not convincing.
  6. In general, quality of figures is not satisfactory at all. They have been carelessly processed. All western blot bands have been cut too close. Figure 1F bands are completely missing. For western blot, total PARP should be immunoblotted and cPARP band should be indicated in the same blot (both bands should be shown). Throughout the manuscript, sometimes Actin has been used as loading control while sometimes GAPDH has been used. They have to be consistent.